# Biofilm Growth on Simulated Fracture Fixation Plates Using a Customized CDC Biofilm Reactor for a Sheep Model of Biofilm-Related Infection

**DOI:** 10.3390/microorganisms10040759

**Published:** 2022-03-31

**Authors:** Walker Kay, Connor Hunt, Lisa Nehring, Brian Barnum, Nicholas Ashton, Dustin Williams

**Affiliations:** 1Department of Orthopaedics, University of Utah, Salt Lake City, UT 84108, USA; kaywalker18@outlook.com (W.K.); connor.h0521@gmail.com (C.H.); lisa.nehring@gmail.com (L.N.); n.ashton@utah.edu (N.A.); 2Purgo Scientific, LLC, South Jordan, UT 84095, USA; brian@purgoscientific.com; 3Department of Biomedical Engineering, University of Utah, Salt Lake City, UT 84112, USA; 4Department of Pathology, University of Utah, Salt Lake City, UT 84112, USA; 5Department of Physical Medicine and Rehabilitation, Uniformed Services University of the Health Sciences, Bethesda, MD 20814, USA

**Keywords:** biofilm, reactor, customized, monomicrobial, polymicrobial, MRSA, *S. aureus*, *P. aeruginosa*

## Abstract

Most animal models of infection utilize planktonic bacteria as initial inocula. However, this may not accurately mimic scenarios where bacteria in the biofilm phenotype contaminate a site at the point of injury. We developed a modified CDC biofilm reactor in which biofilms can be grown on the surface of simulated fracture fixation plates. Multiple reactor runs were performed and demonstrated that monomicrobial biofilms of a clinical strain of methicillin-resistant *Staphylococcus aureus*, *S. aureus* ATCC 6538, and *Pseudomonas aeruginosa* ATCC 27853 consistently developed on fixation plates. We also identified a method by which to successfully grow polymicrobial biofilms of *S. aureus* ATCC 6538 and *P. aeruginosa* ATCC 27853 on fixation plates. This customized reactor can be used to grow biofilms on simulated fracture fixation plates that can be inoculated in animal models of biofilm implant-related infection that, for example, mimic open fracture scenarios. The reactor provides a method for growing biofilms that can be used as initial inocula and potentially improve the testing and development of antibiofilm technologies.

## 1. Introduction

Bacterial biofilm-related infections complicate open fracture sites, thereby challenging clinical care and patients’ quality of life [1,2,3,4]. Since open fracture severities were defined in 1984 [5], rates of infection have remained largely unchanged (ranging from 18–52%) [6,7]. Time to treatment following trauma is directly proportional to the risk of open fracture infection [8]. Orthopedic surgeons continue to seek solutions to manage and treat these infections in both military and civilian sectors. Animal models that more closely replicate contamination events at the point of injury, or that replicate the presence of mature biofilms in an infected site are pivotal to improving the development of therapeutics that can specifically target the biofilm phenotype and accompanying infections [9,10,11,12,13].

Biofilms are dynamic communities of organisms, containing cells in a variety of metabolic states [14,15]; more metabolically active cells dwell in the exterior regions of the community, whereas more quiescent cells with lower metabolism reside deep in the biofilm core. Deeper-residing cells are often referred to as “persisters [14,16]”; due to their lower metabolic state, they are less susceptible to antibiotic perturbation and thus “persist” in wound sites and contribute to recalcitrant biofilm-related infections. Estimates suggest that the majority of bacteria in natural ecosystems—including soil, mud, resident flora of skin, or mucosal membranes—preferentially dwell in the biofilm phenotype [17]. As such, open fractures that are contaminated at the point of injury with soil, mud, or resident flora, are likely to be contaminated with bacteria residing in dynamic biofilm communities [9].

Most animal models that mimic traumatic events or clinical scenarios utilize planktonic (free-floating) bacterial cells as initial inocula to develop an infection signal. The concept is that as planktonic bacteria adhere to hardware in the surgical site, they can form a biofilm and cause infection. Yet, planktonic inocula may not have the same infectious potential or immediate antibiotic recalcitrance as biofilm inocula [9]; we showed in a swine model that wound sites inoculated with planktonic bacteria may be cleared more readily by the host than biofilm inocula [11]. Phagocytic limitations may explain the outcome. Planktonic bacteria have diameters of ~1 µm and thus can be phagocytosed more readily than bacteria in biofilms. Biofilms develop structures in the order of tens to hundreds of µm [18], which frustrate phagocytic clearance as neutrophils cannot phagocytose materials larger than their own diameter (~10 µm) [19].

Planktonic phenotypes are more susceptible to antibiotic therapies than biofilm inocula [20,21]. Consider a clinical scenario wherein antibiotic(s) is administered systemically close to the time of an injury or even at the point of injury. A patient may receive a prophylactic dose of antibiotic that reaches a blood concentration of ~10–50 µg/mL.

If planktonic bacteria are susceptible to the antibiotic(s) (e.g., a minimum inhibitory concentration (MIC) value of 0.5–2 µg/mL depending on the antibiotic), and blood levels are 5–50× in excess of the MIC, the antibiotic(s) will have greater potential to kill the planktonic phenotypes than the recalcitrant fraction of the biofilm; some biofilm phenotypes are known to be 1000× more tolerant to antibiotics than their planktonic counterparts [20] (in our lab we observe biofilms that are 64,000× more tolerant to antibiotics compared to planktonic bacteria). An antibiotic blood level of 500 µg/mL or more may be required to eradicate established biofilms that contaminate an injury site. Yet, such blood levels cannot be achieved due to toxic thresholds.

Thus, therapeutics that are administered and tested against planktonic inocula in animal models may lead to false positive outcomes in cases where biofilm inocula may be more relevant—such as open fracture scenarios. The limitations of systemic antibiotic therapies drive surgeons to consider alternative approaches including directly adding antibiotic powder to surgical sites [22,23,24], or loading antibiotics into cement beads and packing them around hardware in hopes that higher doses of the antibiotic will be delivered and affect the biofilm [25,26,27]. Biofilms drive these approaches as they underpin the recalcitrant nature of difficult-to-treat infections [3].

Functionally, inoculating with planktonic bacteria during an experimental surgery is usually performed by either directly pipetting a bacterial suspension in the experimental wound or dipping implant materials into a bacterial suspension prior to placement. With this technique, it is difficult to keep the inoculation solution in its intended location at its intended concentration. These solutions frequently drip out of the wound as the margins are sewn together upon closure and contaminate non-target sites. Stationary biofilms are more manageable as they stay put and produce exceptionally reproducible infection outcomes [10,11,28].

We have shown that biofilm inocula develop low-grade, chronic infection in sheep models [10,12,28]. This work also involved a modified CDC biofilm reactor in which biofilms were grown on polyetheretherketone (PEEK) membranes [29]. Yet, a limitation of that reactor and inoculation approach is that the PEEK introduces an additional foreign body that is not present when a fracture fixation plate is surgically placed [10,28].

Thus, in this study we designed and developed a customized CDC biofilm reactor to grow biofilms directly on the surface of simulated fracture fixation plates, which builds on our previous work, advances our sheep model of recalcitrant open fracture infection [10,28], and allows for an improvement of antibiofilm therapies. The fixation plates were designed for surgical implantation onto the proximal medial aspect of sheep tibiae; the plates serve as the substrate for biofilm growth and inoculation into a surgical site that mimics the clinical scenario of an open fracture that is contaminated with biofilm at the point of injury, and/or that displays immediate recalcitrance to antibiotic therapies as a model of biofilm implant-related infection.

We quantitatively and qualitatively assessed the reactor’s versatility to growing monomicrobial biofilms of methicillin-resistant *Staphylococcus aureus* (MRSA), *S. aureus* ATCC 6538, *Pseudomonas aeruginosa* ATCC 27853, and polymicrobial biofilms of *S. aureus* ATCC 6538 and *P. aeruginosa* ATCC 27853.

## 2. Methods and Materials

### 2.1. Materials

CDC biofilm reactor glass vessels, hollow tubes for a CDC biofilm reactor lid, baffles, and baffle rods were purchased from Biosurface Technologies (Bozeman, MT, USA). Polypropylene, 316L stainless steel, and Grade 5 titanium (TiV4Al6) stock materials were purchased from McMaster Carr (Elmhurst, IL, USA). Brain heart infusion broth (BHI) was purchased from Becton Dickinson (Franklin Lakes, NJ, USA; via Fisher Scientific (Waltham, MA, USA)). General materials and reagents were purchased from Fisher Scientific. Peristaltic pump tubing, reactor inlet tubing, and effluent tubing were purchased from Cole Parmer (Chicago, IL, USA).

### 2.2. Isolate Selection

*S. aureus* ATCC 6538, and *P. aeruginosa* ATCC 27853 were purchased from the American Type Culture Collection (ATCC; Manassas, VA, USA). *S. aureus* ATCC 6538 is a common isolate used for standardized tests, including military specifications MIL G-13734B. *P. aeruginosa* ATCC 27853 is also a common standard isolate used in susceptibility testing and quality control. MRSA was a clinical isolate collected from a patient who underwent knee surgery and developed an infection; the isolate has been used as an inoculum source in animal models of infection [10,28,30,31].

### 2.3. Simulated Fracture Fixation Plates and Customized CDC Biofilm Reactor

Simulated fracture fixation plates were machined out of Grade 5 titanium. Each plate had dimensions of 1.75 cm × 1.75 cm × 0.85 mm (Figure 1A). Corner radii were machined to 1.25 mm. Four holes with 2.85 mm diameter (distance from plate edge to circle middle was 4.7 mm) were drilled into each plate corner to house cortical bone screws (Figure 1A).

Reactor lids (polypropylene) were custom machined to house holding arms that were designed to suspend the simulated fracture fixation plates in the base of the reactor (Figure 1B). The reactor lid had the same dimensions as a CDC biofilm reactor with the exception of the holding arm ports, which were machined to hold four guillotine-like arms. Each port was oval and had a slight inset such that the arm would seat into it and prevent air transfer to reduce potential contamination (Figure 1B). The lid was machined to accommodate the three hollow tubes that surround the glass baffle rod system of a CDC biofilm reactor. Hollow tubes were press fit and placement-matched those of a standard CDC biofilm reactor. The glass rod and attachment system were likewise the same as a standard CDC biofilm reactor.

Similar to the modified membrane CDC, biofilm reactor we developed previously, whereby each arm had a guillotine-like design; each consisted of an oval head with a rectangular attachment point and a rectangular base that secured two parallel bars (Figure 1C). Each bar of the arm (7.357 in long × 0.25 in wide × 0.25 in deep) was machined with a slot (0.106 in wide × 0.051 in deep) that ran the length of each parallel bar and a notch (0.827 in height) such that two simulated fracture fixation plates could be placed into the notch and slide down the slots of the arm and reside in the base of the reactor vessel (Figure 1C). The customized reactor held a total of eight fracture fixation plates (Figure 1D).

The simulated fracture fixation plates were specifically designed to reside on the proximal medial aspect of sheep tibiae, consistent with previous work (Figure 2).

### 2.4. Monomicrobial Biofilm Growth and Quantification

The customized CDC biofilm reactor and accompanying parts were washed with Alconox detergent and rinsed for 30 min under running de-ionized (DI) or reverse osmosis (RO) water.

Prior to growing biofilms on simulated fracture fixation plates, the plates were passivated and etched to remove machining oil, clean, and roughen the surface to improve biofilm growth. Plates were sonicated for 10 min in Alconox detergent, then rinsed for 10 min in DI or RO water. Plates were placed in 50–100 mL beakers and exposed to a 10:1 ratio of 70% nitric acid:DI water for 30 min. They were again rinsed in running DI or RO water for 10 min, then placed into the arms of a clean reactor. The reactor assembly and plates were autoclaved at 121 °C for a minimum of 20 min.

Biofilms were grown on the plates using a modified protocol from ASTM standard E2562-07 [10,29,32,33]. In short, two to three colonies of an overnight bacterial culture on tryptic soy agar (TSA) or Columbia blood agar were collected and used to adjust bacterial suspensions of a 0.5 McFarland standard (~7.5 × 10^7^ colony forming units (CFU)/mL) using a colorimeter. The reactor was filled aseptically with 500 mL of 100% BHI broth. One mL of a bacterial suspension was aseptically inoculated into the 500 mL of BHI broth, resulting in a concentration of ~1.5 × 10^5^ CFU/mL.

The reactor was placed on a hot stir plate set at 130 rpm and 34 °C. To reduce the effect of temperature fluctuations in the reactor broth (ambient temperatures fluctuated in the lab), Styrofoam was custom cut and placed around the base of the glass vessel of the reactor (Figure 1).

Bacteria were allowed to grow and condition their environment in batch phase (no broth flow) for 24 h. Diluted BHI broth (10%) was then flowed through the reactor for an additional 24 h using a peristaltic pump set at 6.94 mL/min. After 48 h of total growth, simulated fracture fixation plates were sterilely removed from the reactor and the level of biofilm burden on each plate was quantified. Quantification was performed by placing a fixation plate into 10 mL of 10% BHI or phosphate-buffered saline (PBS), vortexing for 1 min, sonicating (47 kHz) for 10 min, vortexing for another 30 s, and using a 10-fold dilution series to plate on tryptic soy agar (TSA) [10,29,31,33,34]. TSA Petri dishes were incubated at 37 °C for 24–48 h and CFU counted to determine CFU/plate following established protocols.

A total of *n* = 25 simulated fracture fixation plates of MRSA (randomly selected) were quantified from 15 separate reactor runs (2–3 plates quantified per reactor; other plates were used as initial inocula in a sheep study not presented here as it is still ongoing). Furthermore, *n* = 30 simulated fracture fixation plates of *S. aureus* ATCC 6538 (randomly selected) were quantified from 19 separate reactor runs (2–3 plates quantified per reactor; other plates were used as initial inocula in the ongoing sheep study). A total of *n* = 25 simulated fracture fixation plates of *P. aeruginosa* ATCC 27853 (randomly selected) were also quantified from 10 separate reactor runs (2–3 plates quantified per reactor; others used in the ongoing sheep study).

### 2.5. Polymicrobial Biofilm Growth and Quantification

The same growth protocol outlined above was used to grow polymicrobial biofilms with the exception of the inoculation procedure. Specifically, *S. aureus* ATCC 6538 and *P. aeruginosa* ATCC 27853 were adjusted to a 0.5 McFarland standard. The *S. aureus* suspension was diluted to 1:1000 and the *P. aeruginosa* suspension was diluted to 1:10,000,000 (we determined and previously published that this ratio difference was required between the two organisms for them to grow together as opposed to one outcompeting the other). Five hundred µL of each bacterial suspension was aseptically added to the 500 mL of broth in the reactor. A timer was set after the first inoculation and another 500 µL of each bacterial suspension was aseptically added to the 500 mL of reactor broth 15 min, 45 min, 1 h 45 min, and 3 h 45 min after the first inoculation. We experimentally determined that five broth inoculations with each isolate improves the development of polymicrobial biofilms.

Furthermore, *n* = 10 plates with polymicrobial biofilms from two separate reactor runs (five plates from each reactor) were likewise quantified as described above.

### 2.6. Imaging

Select fixation plates were photographed after being removed from a reactor to observe general biofilm growth patterns.

A subset of fracture fixation plates were also processed for scanning electron microscopy (SEM) to observe biofilm morphologies on the simulated fracture fixation plate surfaces following biofilm growth. Select fixation plates were aseptically removed from a reactor following the growth protocol, fixed in 10% formalin, and dehydrated in ascending concentrations of ethanol (70%, 95%, 100%) with 3 × 2 h exchanges per concentration. Samples were air dried and sputter coated with gold, then imaged in a JEOL JSM-6610 SEM.

### 2.7. Outlier Handling

Some fixation plates had variable growth, which may have been affected by temperature fluctuations in the lab (cold nights challenge an exposed reactor system). To handle outliers, the mean and standard deviation of each group was calculated. Each sample was given a z-score using the Z = (Mean-sample)/standard deviation equation. Outliers were considered to have a z-score of greater than 3 or less than −3, otherwise known as three standard deviations from the mean. Only one reactor of *P. aeruginosa* ATCC 27853 was removed from the study, all other samples fell within three standard deviations of the mean of their respective groups.

## 3. Results

### 3.1. Monomicrobial Biofilms

MRSA, *S. aureus* ATCC 6538, and *P. aeruginosa* ATCC 27853 biofilms grew without contamination events, which was important to confirm outcomes as monomicrobial biofilms. Quantification data indicated that MRSA produced biofilms with 9.06 ± 0.44 log_10_ CFU/plate (Figure 3). *S. aureus* ATCC 6538 produced biofilms with 8.56 ± 0.90 log_10_ CFU/plate (Figure 3). *P. aeruginosa* produced biofilms with 8.92 ± 0.22 log_10_ CFU/plate (Figure 3). Overall, each isolate produced a CFU/plate count that was within ~0.5 log_10_ units.

Photographs showed that biofilms of each isolate on fixation plates were visible with the naked eye. Biofilm appeared to accumulate to a greater degree near the edged regions of each fixation plate, such as the outer border or cliff regions of drill holes (Figure 4A). Nevertheless, the biofilm coverage was extensive across the surfaces of all plates. SEM images likewise confirmed a relatively even distribution of biofilm growth across fixation plate surfaces (Figure 4).

SEM images of monomicrobial MRSA biofilms showed that they produced three-dimensional structures, similar to what we observed previously with the same isolate. Indications of water channels and extracellular polysaccharide substance (EPS), or biofilm matrix components were readily observable (Figure 4C). MRSA biofilms produced classic plume-like structures (Figure 4B). Bacterial cells appeared to preferentially accumulate within or around a three-dimensional structure as opposed to residing individually on the surface of the plate (Figure 4B). A similar morphology and pattern was observed for *S. aureus* ATCC 6538, yet biofilm communities of this isolate tapered to a point more so than creating a plateau as MRSA did (Figure 4D).

We observed the biofilm structure of *P. aeruginosa* ATCC 27853 previously [33]. Data corroborated these previous observations; *P. aeruginosa* biofilms produced more sheet-like biofilm structures than plume-like formations. Yet, subtle regions of raised formations were observed (Figure 4).

SEM images also showed that there was a distinct linear region along the outer portions of the plates where biofilms formed to a lesser degree. This region is where simulated fracture fixation plates came in contact with the slide track (slot regions) of the parallel bars of the holding arm (Figure 4). The same pattern was observed for each isolate.

### 3.2. Polymicrobial Biofilms

Similar observations as those made with monomicrobial biofilm formation were seen with polymicrobial biofilms; they distributed relatively evenly across the surface of simulated fracture fixation plates (Figure 5A) and were visible to the naked eye. Yet, gross photographs or low-resolution SEM images were not able to indicate whether *S. aureus* ATCC 6538 and *P. aeruginosa* ATCC 27853 successfully developed polymicrobial biofilms. Higher resolution SEM images were required to determine the isolates’ dual growth properties.

SEM data showed that the two isolates successfully developed polymicrobial biofilms. Indeed, the polymicrobial biofilm communities were robust with intriguing cotton ball-like formations, several of which developed bridged connections between plumes (Figure 5B). Interestingly, whereas *P. aeruginosa* ATCC 27853 developed sheet-like structures as a monomicrobial biofilms, the isolate appeared to not only adopt, but enhance the development of three-dimensional biofilm structures when combined with *S. aureus* ATCC 6538 (Figure 5B–D). Overall, the two cell types appeared to integrate seamlessly into a biofilm community with EPS, or matrix components that were readily produced and observable (Figure 5C,D).

Because these isolates produce different morphologies on TSA, we were able to identify and count individual colonies and calculate the respective CFU/plate for each isolate. Quantification data indicated that there were more *S. aureus* ATCC 6538 cells in the polymicrobial biofilms than *P. aeruginosa* ATCC 27853. Specifically, there were approximately 9.81 ± 0.10 log_10_ CFU/plate for *S. aureus* and 8.17 ± 0.05 log_10_ CFU/plate for *P. aeruginosa* (Figure 3). Although the isolates were inoculated into BHI broth with a 4 log_10_ difference between the inoculation amounts, there was only a ~2 log_10_ difference in CFU counts between isolates following growth (Figure 3).

*P. aeruginosa* appeared to enhance *S. aureus* growth compared to when it was grown as a monomicrobial biofilm, whereas *P. aeruginosa* had slightly fewer CFU/plate compared to its monomicrobial counterpart. However, there was no statistically significant difference in any comparison between groups (*p* < 0.05 for all following an independent samples *t* test).

Taken together, the data indicated that the customized CDC biofilm reactor was able to successfully produce monomicrobial and polymicrobial biofilms on the surface of simulated fracture fixation plates.

## 4. Discussion

Biofilms underpin difficult-to-treat implant-related infections, which adversely affect open fractures and accompanying hardware [3,13]. Improved antibiofilm therapies are still in demand. We propose that one of the most promising improvements that can be made toward translating effective antibiofilm therapies is the use of biofilms as initial inocula in animal models of infection [9].

To date, biofilm reactor systems have primarily been developed for in vitro analyses (e.g., CDC biofilm reactor, drip flow reactor, rotating disk biofilm reactor) not for in vivo inoculation procedures and subsequent analysis. We have developed multiple reactor systems that address this limitation. For example, we developed a reactor to grow biofilms on PEEK membranes for inoculation into the proximal medial aspect of sheep tibiae [10,28], another to grow biofilms on collagen coupons for inoculation into excision wounds of a pig back model [11], and another (developing) to grow biofilms on the surface of silica beads that model sand particles that contaminate blast-related injuries [12].

The customized CDC biofilm reactor presented herein was developed to contribute to these efforts and provide a method by which to inoculate a traumatic injury site with biofilms grown directly on the surface of simulated fracture fixation plates; thereby improving upon our sheep model of traumatic and instantly recalcitrant open fracture-related infection. The sheep model is being used to determine efficacy profiles of various antibiofilm technologies that are on the market and that are under development.

Notably, biofilm growth methods and inoculation approaches are not limited to sheep. We are currently developing biofilm reactor systems and inoculation methods for the knee joint space in rats. We also envision additional reactor growth systems with varying substrates including glass, collagen, or metal that can be scaled for rabbit or mice models.

Data indicated that monomicrobial biofilms of MRSA, *S. aureus* ATCC 6538, and *P. aeruginosa* ATCC 27853 had similar CFU/plate, which was promising as the reactor system and growth protocol resulted in consistent growth across multiple isolates. Quantification data were pooled for each isolate across the various reactor runs, which meant that robustness and repeatability were not strictly determined for each individual run yet indicated that consistent growth could be achieved after multiple reactor runs.

Our quantification procedure for polymicrobial biofilms resulted in growing *S. aureus* ATCC 6538 and *P. aeruginosa* ATCC 27853 on the same agar plates following vortexing and sonicating. Colony morphology differences were used to calculate the CFU. As these isolates are known to have competitive effects, future work may be needed to confirm that *S. aureus* does in fact produce higher counts than *P. aeruginosa*; differential agars may more accurately determine their growth profiles. One reason we mention this is, as we observed the SEM images, there appeared to be more *P. aeruginosa* cells than *S. aureus*, yet SEM images are limited in that they provide an analysis of the surface of the biofilm structure; there could have been more *S. aureus* cells below the outer surface, which led to increased counts via quantification.

We used stainless steel to machine the holding arms and titanium to machine the plates. Different materials could have been used and can be customized based on need or desire. Yet, because the plate materials come in contact with the holding arms, which was made obvious by the reduced biofilm growth in those regions where the materials came in contact, it is important to consider material selection and potential interaction. Various metals interact with one another in aqueous environments. Data indicate that stainless steel–titanium interactions may be less of a problem than stainless steel–stainless steel interactions. We have not yet tested biofilm growth on stainless steel plates in this particular reactor system, yet if other materials were to be selected, interaction effects should be considered.

## 5. Conclusions

The machining, assembly, and growth protocol for a customized CDC biofilm reactor was straightforward and achievable. This reactor system or modifications thereof have the potential to broaden the selection profile of reactors that can be used to grow biofilms and can be used as initial inocula in animal models to produce recalcitrant, hardware-related infections, and improve the development of antibiofilm therapies.

## Figures and Tables

**Figure 1 microorganisms-10-00759-f001:**
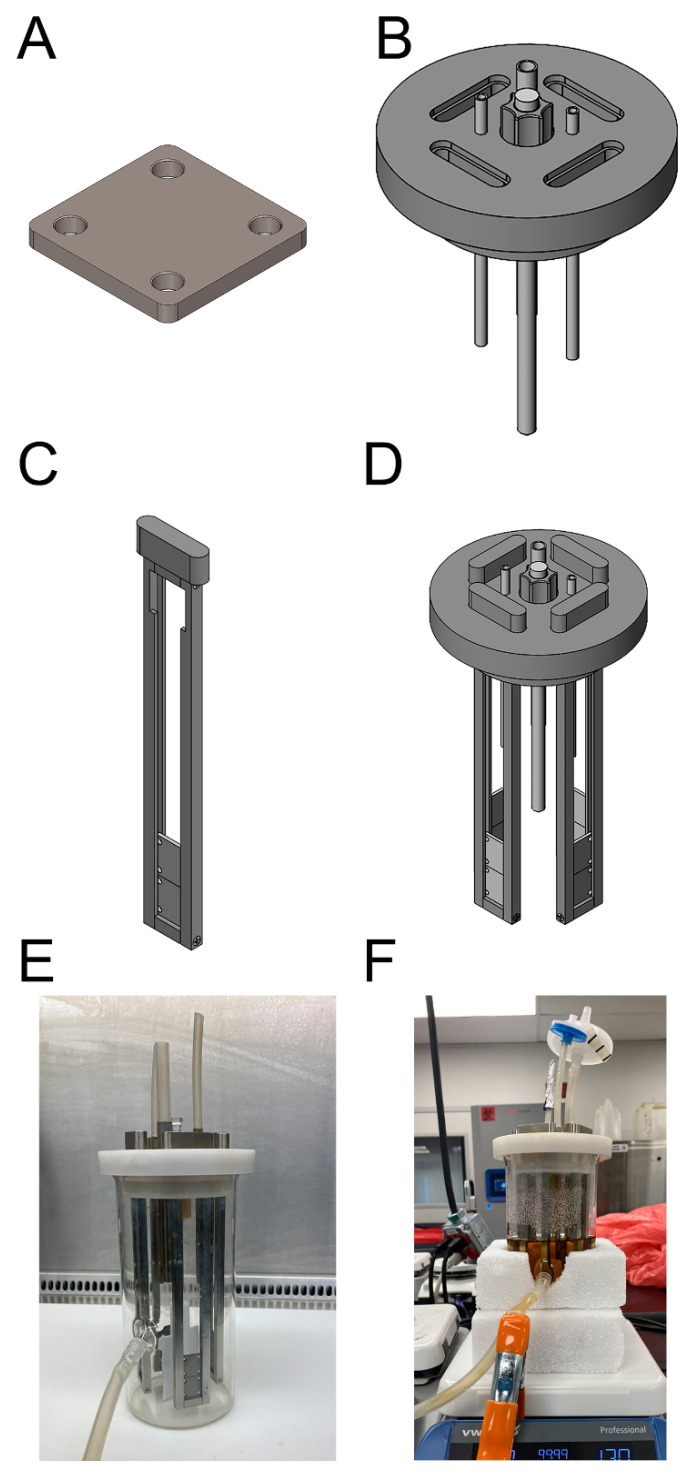
Customized CDC biofilm reactor. (**A**) Model of a simulated fracture fixation plate. (**B**) Model of the customized lid with oval openings through which reactor holding arms can be inserted and hold the fixation plates. (**C**) Model of a holding arm into which fixation plates can be placed. (**D**) Model of the reactor lid and holding arms, each with two fixation plates (total of *n* = 8 plates/reactor). (**E**) Assembled reactor with relevant tubing consistent for reactor use. (**F**) Reactor with “cozies” surrounding the bottom portion; cozies reduced temperature fluctuations due to temperature swings in the lab.

**Figure 2 microorganisms-10-00759-f002:**
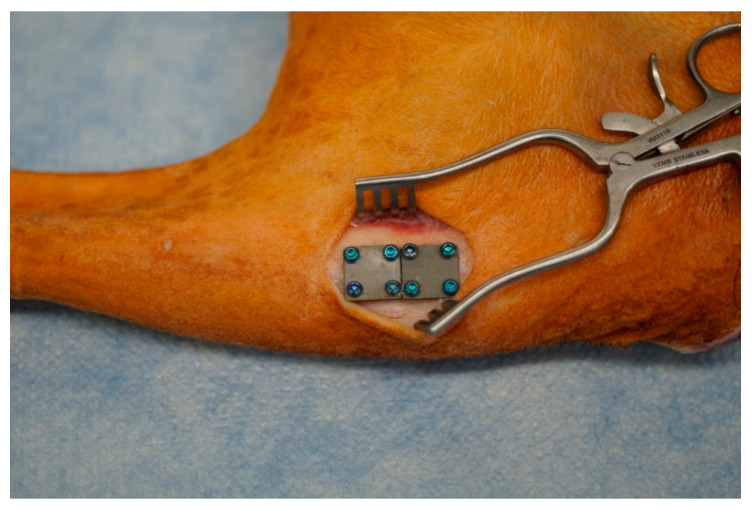
Two simulated fracture fixation plates secured to the proximal medial aspect of a cadaveric sheep tibia (from a separate IACUC-approved study). This region of sheep bone is relatively flat. Plates are sized to reside (mostly) in plane with the flat portion of the bone.

**Figure 3 microorganisms-10-00759-f003:**
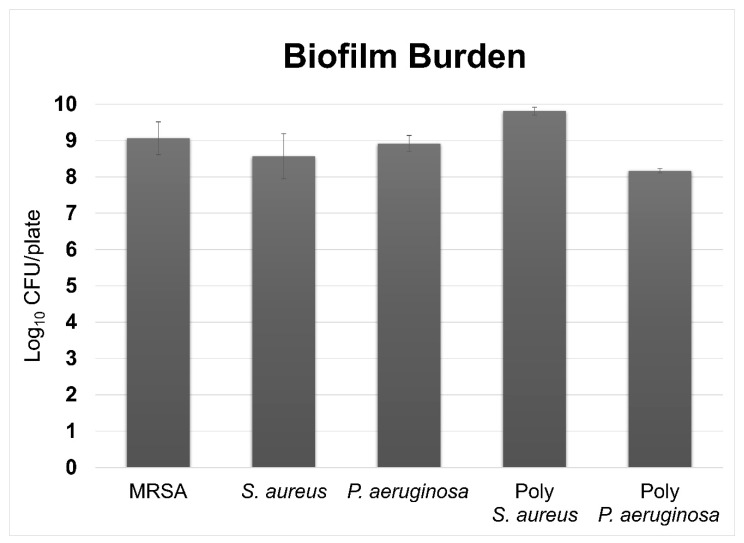
Quantification results for monomicrobial and polymicrobial biofilms. MRSA, *S. aureus* ATCC 6538, and *P. aeruginosa* ATCC 27853 had similar bioburden levels as monomicrobial biofilms, each having close to 10^9^ CFU/plate. *S. aureus* ATCC 6538 had roughly 10^2^ more CFU/plate than *P. aeruginosa* ATCC 27853 in the polymicrobial biofilms. *S. aureus* ATCC 6538 in the polymicrobial biofilms had ~0.8 log_10_ more CFU than its monomicrobial counterpart, whereas *P. aeruginosa* ATCC 27853 had ~0.8 log_10_ less CFU than its monomicrobial counterpart.

**Figure 4 microorganisms-10-00759-f004:**
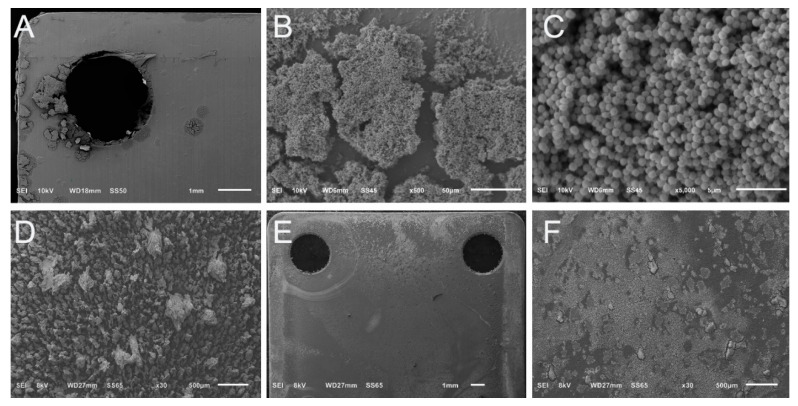
Representative SEM images of monomicrobial biofilm growth. (**A**) Stitched image collated from multiple high resolution micro graphs showing the growth pattern of MRSA biofilms on a simulated fracture fixation plate. Biofilm growth was more prominent along the border and screw hole regions. The distinct line that runs along the top of the plate indicates the region where the plate was in the slot of the reactor holding arm. (**B**) Higher resolution image of MRSA biofilms indicate that the biofilm colonies had a flat, plateau-like top. (**C**) Representative image of MRSA biofilms showing the presence of EPS materials. (**D**) Biofilm morphology of *S. aureus* ATCC 6538. Biofilm communities had significant three-dimensional structures that tapered to a point, in contrast to MRSA biofilms which had a plateau-like appearance. (**E**) Stitched image of *P aeruginosa* ATCC 27853 biofilms on the surface of a simulated fracture fixation plate. Image was collated from multiple high-resolution images. Similar to what was observed for each isolate, biofilms showed relatively uniform coverage across the surface of the simulated fracture fixation plate. (**F**) Higher resolution image of *P aeruginosa* ATCC 27853 biofilms. This isolate produced sheet-like structures of biofilms as opposed to the distinct plumes of *S*. *aureus*.

**Figure 5 microorganisms-10-00759-f005:**
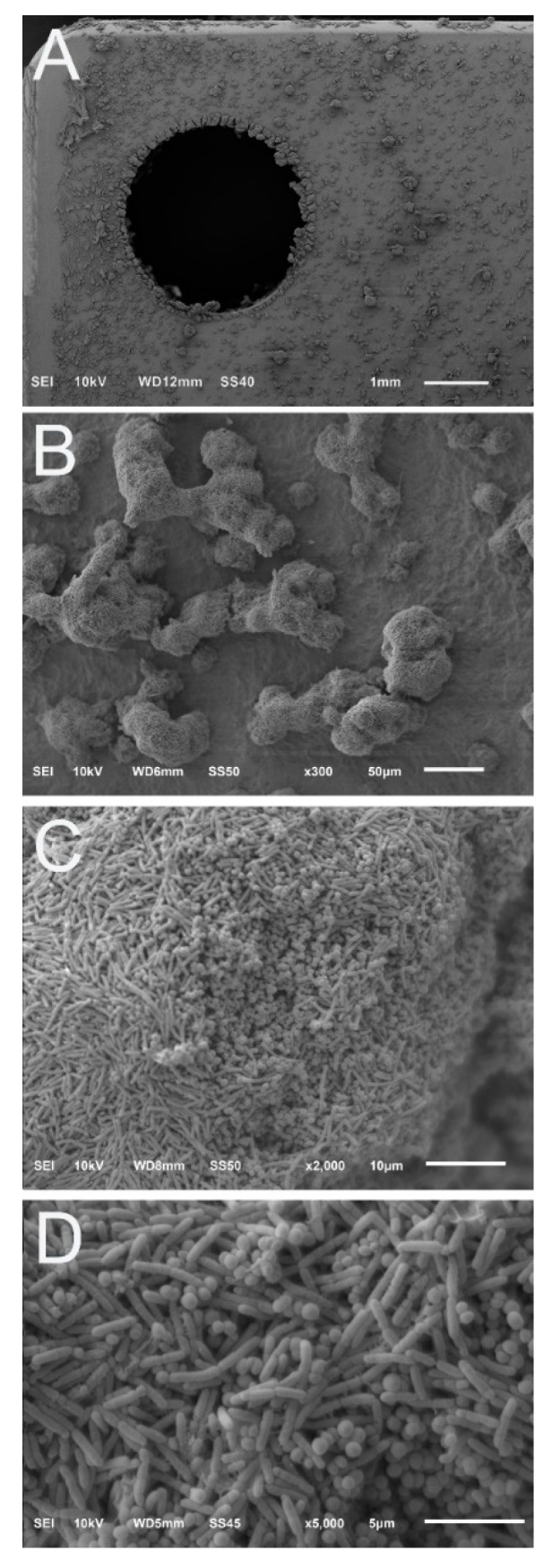
Representative SEM images of polymicrobial biofilms of *S*. *aureus* ATCC 6538 and *P*. *aeruginosa* ATCC 27853. (**A**) Stitched image of multiple high-resolution SEM images of polymicrobial biofilms. The biofilm plumes were noticeably more distinct and larger than those of the monomicrobial biofilms. (**B**) Biofilm communities had three-dimensional connected cotton ball-like morphologies that did not resemble their monomicrobial biofilm morphologies (compare Figure 3D–F). (**C**,**D**) Higher resolution images of the polymicrobial biofilms indicated that both cell types were present and appeared to integrate seamlessly with EPS material present.

## Data Availability

The data presented in this study are available within the article.

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
