# Peer review of "Biofilm Growth on Simulated Fracture Fixation Plates Using a Customized CDC Biofilm Reactor for a Sheep Model of Biofilm-Related Infection"

_microorganisms, 2022, doi:10.3390/microorganisms10040759_

Round 1

Reviewer 1 Report

Overall good development. One issue that needs to be explained as a caveat is the nature of biofilm prepared in this reactor vs. biofilm that develops in vivo over time, especially if exposed to antibiotics. The reactor produces a biofilm with a very large amount of cells, but it is unlikely that all of those cells are in the dormant, persistent state. In reality, only a small portion of the cells may be in the difficult to kill state. As a result, the number of cells reported as a starting inoculum can be misleading, because it sounds like there are 10^9 hard to kill biofilm cells when in reality, there may be 10^2 or 10^3 really hard to kill cells with 10^6 cells that are more metabolically active. There needs to be further characterization of the biofilm metabolic heterogeneity and phenotypic heterogeneity so that a more accurate estimation can be obtained. In addition, use of the reactor in a "buildup biofilm" approach with successive antibiotic treatment may generate more clinically realistic challenging biofilms. 

Reviewer 2 Report

This is an interesting proof of concept study in which authors validate a new method to inoculate a traumatic injury site with biofilms grown directly on the surface of simulated fracture fixation plates.

My only concern is that they use it for a sheep model. However, it is not a very common animal for in vivo models. Could their customized CDC biofilm reactor be used with rabbits or mice? I consider they should discuss it in the discussion section.

They should mention that it is a “proof of concept” study.

Minor comments:

  1. In lines 16 and 90 I should better say: “biofilm reactor where biofilms” or substitute “with which” by other expression, as it appears now sounds strange.
  2. Abstract and methods: specify that the methicillin-resistant aureus is a “clinical strain”. And clarify whether the S. aureus ATCC6538 is a methicillin-susceptible strain.
  3. Methods: Do they authors previously tested the biofilm production of the strains used? i.e. using crystal violet assay?
  4. Figure 3. Put genus and species names in italics.
